# Profile Characterization of Biogenic Amines in Glioblastoma Patients Undergoing Standard-of-Care Treatment

**DOI:** 10.3390/biomedicines11082261

**Published:** 2023-08-13

**Authors:** Orwa Aboud, Yin Liu, Lina Dahabiyeh, Ahmad Abuaisheh, Fangzhou Li, John Paul Aboubechara, Jonathan Riess, Orin Bloch, Rawad Hodeify, Ilias Tagkopoulos, Oliver Fiehn

**Affiliations:** 1Department of Neurology, University of California, Davis, Sacramento, CA 95817, USA; 2Department of Neurological Surgery, University of California, Davis, Sacramento, CA 95817, USA; 3Comprehensive Cancer Center, University of California Davis, Sacramento, CA 95817, USA; 4Department of Ophthalmology, University of California, Davis, Sacramento, CA 95817, USA; 5West Coast Metabolomics Center, University of California Davis, Davis, CA 95616, USA; 6Department of Pharmaceutical Sciences, School of Pharmacy, The University of Jordan, Amman 11942, Jordan; 7School of Medicine, Al Balqa Applied University, Al-Salt 19117, Jordan; 8Department of Computer Science, University of California, Davis, Sacramento, CA 95616, USA; 9Genome Center, University of California, Davis, Sacramento, CA 95616, USA; 10USDA/NSF AI Institute for Next Generation Food Systems (AIFS), Davis, CA 95616, USA; 11Department of Internal Medicine, Division of Hematology and Oncology, University of California, Davis, Sacramento, CA 95817, USA; 12Department of Biotechnology, School of Arts and Sciences, American University of Ras Al Khaimah, Ras Al-Khaimah 10021, United Arab Emirates

**Keywords:** biogenic amines, metabolomic profiling, glioblastoma, concurrent chemoradiation

## Abstract

Introduction: Biogenic amines play important roles throughout cellular metabolism. This study explores a role of biogenic amines in glioblastoma pathogenesis. Here, we characterize the plasma levels of biogenic amines in glioblastoma patients undergoing standard-of-care treatment. Methods: We examined 138 plasma samples from 36 patients with isocitrate dehydrogenase (IDH) wild-type glioblastoma at multiple stages of treatment. Untargeted gas chromatography–time of flight mass spectrometry (GC-TOF MS) was used to measure metabolite levels. Machine learning approaches were then used to develop a predictive tool based on these datasets. Results: Surgery was associated with increased levels of 12 metabolites and decreased levels of 11 metabolites. Chemoradiation was associated with increased levels of three metabolites and decreased levels of three other metabolites. Ensemble learning models, specifically random forest (RF) and AdaBoost (AB), accurately classified treatment phases with high accuracy (RF: 0.81 ± 0.04, AB: 0.78 ± 0.05). The metabolites sorbitol and N-methylisoleucine were identified as important predictive features and confirmed via SHAP. Conclusion: To our knowledge, this is the first study to describe plasma biogenic amine signatures throughout the treatment of patients with glioblastoma. A larger study is needed to confirm these results with hopes of developing a diagnostic algorithm.

## 1. Introduction

Glioblastoma is the most common primary malignant brain tumor in adults and universally carries a poor prognosis [1]. First-line treatments typically include maximal safe resection, radiation therapy (RT), chemotherapy using temozolomide (TMZ), and tumor treating field (TTF) therapy. This treatment is associated with improvement in 2-year survival rates compared with prior treatments; however, the glioblastoma inevitably recurs, and the survival rate at 5 years is a dismal 6.6% [2,3]. These tumors acquire new mutations that result in resistance to treatment [4]. Further, these mutations can lead to changes in plasma metabolites that are detectable via metabolomic techniques at the time of diagnosis [5]. We have recently reported changes in metabolomic profiles in patients undergoing treatment [6]. However, there has not been a close examination of the metabolite class of biogenic amines in patients undergoing treatment for glioblastoma.

Biogenic amines are an important class of metabolites as they play critical roles in cell metabolism, cell growth, and intracellular communication [7]. They are defined as nitrogenous organic compounds—primary (R-NH_2_), secondary (R_2_-NH), or tertiary (R_3_-N) amine structures—with known physiologic effects [8,8]. In the nervous system, they function in numerous processes, including protein synthesis and neurotransmission. Furthermore, several studies have implicated various amines in glioma pathogenesis. Elevated glycine levels are associated with cancer cell survival [9]. Glutamate release by glioma cells has been implicated in the aggressive nature of the disease [10].

In this prospective study, we enrolled a cohort of patients with isocitrate dehydrogenase (IDH) wild-type glioblastoma, as defined by the new World Health Organization (WHO) classification and performed untargeted metabolomics of the patients’ plasma, with a focus on biogenic amines before and after surgery, as well as before and after concurrent chemoradiation. Similar approaches have previously been carried out on studies of patients with other types of cancers, including hepatobiliary and endometrial cancer [11,12,13]. We hypothesized that biogenic amines play an important role in glioma pathogenesis, and as such, we identified significant changes in their levels throughout different stages of treatment. We also propose a data-driven approach of integrating metabolomics with machine learning to categorize plasma levels of biogenic amines in glioblastoma patients with the goal of developing a predictive tool for the diagnosis and prognosis of this disease.

## 2. Materials and Methods

### 2.1. Patients

Thirty-six patients with histopathological confirmed diagnoses of IDH wild-type glioblastoma, WHO grade 4 [14], were enrolled into this study. Approval from the Institutional Review Board of The University of California Davis was obtained along with written informed consents from all patients. We obtained demographic and clinical data for the study subjects via medical record review. Patients received standard-of-care initial treatment with surgical resection, concurrent radiation therapy, and chemotherapy. Blood samples were collected before surgery (BS), two days after surgery (S), prior to starting radiation therapy (RT), after completing radiation therapy (PRT), as well as after adjuvant chemotherapy treatment (PT). Of note, none of these patients received TTF therapy during the time of our sample collection.

### 2.2. Biogenic Amines Profiling

Untargeted plasma metabolomics via hydrophilic interaction liquid chromatography triple time of flight TTOF mass spectrometry (HILIC-TTOF MS) was performed at the UC Davis West Coast Metabolomics Center. Plasma metabolite extraction has been previously described [15]. Metabolite profiling using HILIC-TTOF-MS was performed on the Agilent 1290 UHPLC/Sciex TripleTOF 6600 mass spectrometer under positive and negative ionization modes. Metabolites (injection volume 5 µL) were separated using a Waters Acquity UPLC BEH Amide column (1.7 µm, 2.1 × 150 mm), and a binary mobile phase consisted of 100% LC-MS grade H_2_O with 10 mM ammonium formate and 0.125% formic acid as solvent A and 95:5 (*v*/*v*) ACN:H_2_O with 10 mM ammonium formate with 0.125% formic acid as solvent B. The mobile phase was running under gradient conditions with a flow rate of 0.8 mL/min. The column temperature was kept at 45 °C. Data were acquired in data-dependent acquisition mode with a mass range 50–1500 *m*/*z* for MS1 and 40–1000 *m*/*z* for MS2.

### 2.3. Raw Data Processing, Metabolite, Annotation, and Statistical Analysis

MS-Dial 4.6 was used to process the acquired raw LC–MS data, including peak-picking, peak alignment, and the annotation of related peaks [15]. Simca P+14 (Sartorius Stedim Data Analytics AB, Umea, Sweden) was used to generate partial least square-discriminative analysis (PLS-DA) score plots. The imported datasets identified metabolite (Rt, m/z) pairs with their normalized peak heights which were mean centered and auto scaled. Of note, although this metabolomic workflow aimed to quantify biogenic amines, it also identified some related metabolites, which were not amines but were related upstream or downstream molecules in cellular metabolism. MetaboAnalyst 5.0 (McGill University, Montreal, QC, Canada) (http://www.metaboanalyst.ca (accessed on 6 August 2023)) [16] was used to generate heat maps. GraphPad Prism 9 (version 9.5, San Diego, CA, USA) was used to generate volcano plots. The processed peak heights with their annotations were imported to MetaboAnalyst, normalized to the total sample median, and auto scaled. Unpaired Student’s *t*-Test was used to identify significantly altered metabolites between the compared groups (*p*-value < 0.05 was considered significant). Enrichment analysis was carried out using the freely available statistical tool ChemRICH (Chemrich.fiehnlab.ucdavis.edu; accessed on 20 June 2023). Individual metabolite abundance comparisons were performed using GraphPad Prism 9 (version 9.5, San Diego, CA, USA).

### 2.4. Machine Learning Modeling: Data Pre-Processing and Machine Learning Models

#### 2.4.1. Dataset

The full dataset consisted of 137 samples after removing one with more than 50% missing biogenic amines. Each sample was associated with one of the five treatment stage classes: pre-surgery, post-surgery, pre-radiation, post-radiation, and post-adjuvant treatment. While splitting the dataset into training and testing sets, we ensured each split was stratified, maintaining the original class distribution. In addition, since multiple samples could be associated with a single patient, we also ensured that samples from the same patient would not appear across different splits. Finally, we used an 80-to-20 split, resulting in 110 and 27 samples for the training and testing sets, respectively (Appendix A).

#### 2.4.2. Model Selection

We implemented a model selection with Scikit-learn [17] to choose the optimal machine learning model for this dataset, where we considered five classifiers: Logistic Regression (LR) [18], Support Vector Machine (SVM) [19], random forest (RF) [20], AdaBoost (AB) [21], and Multilayer Perceptron (MLP) [22]. For each classifier, we performed a grid search with 5-fold cross-validation to tune the best hyperparameter (Appendix A). Note that while generating splits for the cross-validation, we ensured that each split was stratified and did not share the same patient, similar to the training–testing splits (Appendix A). Finally, for the selected model with the best accuracy, we performed sequential feature selection implemented by MLxtend [23] to select the most predictive subset of features.

## 3. Results

### 3.1. Patients

As we reported in our recent publication [6], thirty-six patients with glioblastoma, IDH wild-type (determined via immunohistochemistry), WHO grade 4, were identified in the UC Davis Neuro-Oncology clinic. As MGMT methylation reflects a good prognostic factor [24], we determined that of the 36 patients, 18 had MGMT promoter methylation, 16 were unmethylated, and 2 had unknown methylation statuses. The cohort consisted of 15 females and 21 males; the median age at diagnosis was 63.5 years; the median BMI at diagnosis was 28; and 30 patients were self-identified as white (Table 1). All patients underwent surgical intervention at our institution. We collected 36 samples before surgery (BS), 32 samples 2 days after (S), 28 samples just before radiation therapy (RT), 17 samples after the radiation therapy had been completed (PRT), as well as 13 samples after completion of adjuvant chemotherapy (PT).

### 3.2. Biogenic Amines

We identified 340 unique biogenic amines via retention index and mass spectral matching [25] (Appendix A). We compared samples at four points in each patient’s treatment course as follows: the first time point was prior to surgery (BS); the second was post-surgery (S); the third was prior to starting concurrent chemoradiation therapy (RT), after finishing concurrent chemoradiation (PRT); and the last time point was after the completion of adjuvant chemotherapy (PT).

#### 3.2.1. Post-Surgery versus Pre-Surgery

In comparing the profile of biogenic amines post-surgery with pre-surgery, partial least squares–discriminant analysis (PLS-DA) showed two-group clustering with evidence of separation (Figure 1D). The complete list of significantly altered biogenic amines is available (Appendix A). Figure 1B illustrates the 25 metabolites with the highest VIP scores. Figure 1A depicts a heat map of the top 40 significantly altered metabolites including biogenic amines. Setting the fold change (FC) cutoff at 2 and *p* value at 0.05 (Figure 1C), there were 12 compounds that were upregulated and 11 compounds that were downregulated (Table 2).

#### 3.2.2. Post-Radiation versus Pre-Radiation

When comparing the profiles of biogenic amines post-radiation with pre-radiation, PLS-DA showed two-group clustering (Figure 2D). Figure 2B illustrates the 25 metabolites with the highest VIP scores. Figure 1A depicts a heat map of the top 40 significantly altered metabolites including biogenic amines. The complete list of significantly altered biogenic amines is available (Appendix A). By setting the FC cutoff at 2 and *p* value at 0.05 (Figure 2C), only three compounds were upregulated and three compounds were downregulated (Table 3). Figure 2C depicts a heat map of the top 40 significantly altered biogenic amines.

#### 3.2.3. Post-Treatment versus Pre-Radiation

Moreover, the profiles of biogenic amines post-treatment and pre-radiation were compared; PLS-DA showed two-group clustering with separation noted (Figure 3D). The complete list of significantly altered metabolites including biogenic amines is available (Appendix A). Figure 3B illustrates the 25 metabolites with the highest VIP scores. Figure 3A depicts a heat map of the top 40 significantly altered metabolites including biogenic amines. Using the same setting of the FC cutoff at 2 and *p* value at 0.05 (Figure 3C), there were seven compounds that were upregulated and eight compounds that were downregulated (Table 4).

### 3.3. Machine Learning Models for Classifying Treatment Phases

#### 3.3.1. Ensemble Learning Accurately Predicted Patient Treatment Stages

Different classifiers with the best-performing hyperparameters are shown in Figure 4. Ensemble learning classifiers, RF and AB, achieved the highest accuracy (0.81 ± 0.04 and 0.78 ± 0.05) and were significantly more accurate than the third best model, SVM (0.69 ± 0.14; *p*-value = 5.9 × 10^−5^ and 2.9 × 10^−3^, respectively). Using RF with the optimal hyperparameters (Appendix A), we evaluated its performance using the holdout test set. The test accuracy and F1 score were both 0.81 ± 0.03 (Figure 5A). We also examined the area under the precision recall curve (AUPRC) and the area under the receiver operating characteristics (AUROC), which enabled an estimation of the predictive validity of the models. The overall AUPRC and AUROC for the combined datasets were 0.87 and 0.96, respectively (Figure 5B–D). Specifically, the model could distinguish pre-surgery (AUPRC = 1.0, AUROC = 1.0) and post-surgery (AUPRC = 0.99, AUROC = 1.0) plasma samples nearly perfectly while somewhat struggling to classify pre-radiation (AUPRC = 0.73, AUROC = 0.87) and post-radiation (AUPRC = 0.71, AUROC = 0.92) patients. Sequential feature selection selected the 25 (7%) most predictive biogenic amines from the original 340, slightly improving the micro-averaged test accuracy and F1 scores (both 0.82 ± 0.02; Appendix A).

#### 3.3.2. Sorbitol and N-Methylisoleucine Were among the Most Predictive Biogenic Amines

We applied multiple methods to rank the most predictive metabolites including biogenic amines (Appendix A). We identified eight metabolites ranked among the top 10 most important features simultaneously using at least two methods (Figure 6A), which were bupivacaine, creatine, galactosamine, linoleic acid, mannitol, N-methylisoleucine, nudifloramide, and sorbitol. To better understand how these metabolites affected the classification of patient treatment stages, we applied a model interpretability method called SHAP [26] to our random forest classifier trained with selected features. Again, sorbitol and N-methylisoleucine were the two most important features (mean|SHAP| = 0.36 and 0.22), whereas galactosamine and nudifloramide were sixth and ninth, respectively (Figure 6B).

## 4. Discussion

In this investigation, distinct alterations in plasma metabolites linked to treatment response were discovered in patients diagnosed with pathologically confirmed IDH wild-type glioblastoma. The primary therapeutic approach for glioblastoma patients encompasses maximal safe resection, radiation therapy, and chemotherapy [2]. Patients with brain tumors detected through imaging typically undergo surgical intervention to obtain tumor tissue for histological diagnosis and molecular testing. Additionally, the degree of resection significantly impacts prognosis. Despite potential molecular variations in glioblastomas among patients, a mostly uniform treatment approach has been implemented for the majority, primarily due to the limited availability of effective and targeted therapies. Metabolomics provides us a powerful tool to examine plasma metabolites in patients with glioblastoma to potentially identify clinically relevant biomarkers. In particular, little is known about how biogenic amines levels change during early treatment stages like surgery and chemoradiation.

In our prospective study, we conducted a comparative analysis of plasma metabolomic profiles collected from patients with IDH wild-type glioblastoma. We examined samples both before and after surgical resection, prior to and after concurrent chemoradiation, and following adjuvant chemotherapy, in accordance with the standard-of-care protocol [2]. Obtaining a metabolomic signature for patients may lead to improved diagnostics and treatment strategies. Interestingly, although this study was primarily aimed at exploring biogenic amines, some of the most significantly altered metabolites are not amines. Regardless, these findings will prove useful for future efforts in developing diagnostic and prognostic metabolite biomarkers.

Glycine is a nonessential amino acid that belongs to the super class of organic acids. Glycine is biosynthesized in the body from amino acid serine and is involved in the body’s production of collagen, hemoglobin, and DNA with the principal function as a precursor to proteins. Glycine and serine both play a role in one carbon metabolism [27]. Glycine is an inhibitory neurotransmitter in the central nervous system, and this might explain our observation of the upregulation of glycine after surgery. Some studies showed elevated glycine after severe traumatic brain injury, and such an upregulation is thought to be a compensatory mechanism to counteract the excitotoxic impacts of glutamate [28].

Of note, elevated serum levels of glycine and serine were linked to a protective role in pancreatic cancer [29]. Kim et al. [9] reported an important role for both serine and glycine in glioblastoma cells’ survival in the “ischemic zones” of gliomas. Betonicine was another amino acid that we noticed the upregulation of after surgery in our samples, though its relevance is unclear.

After surgery, certain external substances called exogenous metabolites could be detected. These include p-acetamidophenyl-beta-D-glucuronide, which is a major urinary metabolite of acetaminophen, and 3-(cystein-S-yl) acetaminophen, an amino acid with organic acids. Another detected metabolite was acetaminophen sulfate, belonging to the phenylsulfate group of organic acids. Additionally, S-methyl-3-thioacetaminophen, which is also a metabolite of acetaminophen, was found. These findings are likely linked to the direct administration of medication right after surgery. Similarly, the levels of 2-hydroxy-5-sulfopyridine-3-carboxylic acid, dehydrofelodipine, and 2-amino-3-methoxybenzoic acid increased after surgery, and the last compound’s levels decreased after treatment completion. This suggests that these compounds may originate from external sources.

Our results included the downregulation of multiple metabolites after surgery; this included metabolites that are related to medications and/or supplements used during and shortly after surgery, such as sugar alcohols (including mannitol, sorbitol, and lactitol) with little significant energy value as they are largely eliminated from the body. Bupivacaine, a long-acting local anesthetic, is only found in individuals that have used or taken this drug. Lastly, 1-hydroxymidazolam-beta-D-glucuronide is the glucuronidated conjugate of a midazolam metabolite.

Lipid metabolism correlates with cell replication; so, decreased levels of linoleic acid may suggest altered cell replication. 1-methylnicotinamide and nudifloramide are pyridine alkaloids. 1-methylnicotinamide was previously reported to be enriched in T cells infiltrating serous carcinoma [30]. Fatty acyls such as hexadecanedioic acid and 3-hydroxybutyric acid were also downregulated after surgery. The upregulation of sterol lipids including glycodeoxycholic acid, glycocholic acid, and taurocholic acid after surgery could be related, at least in part, to the dysregulated lipid metabolism in glioma [31]. The downregulation of riluzole, a benzothiazole with an organoheterocyclic compound super class, is not fully clear, as our patients were not on exogenous treatment with riluzole. This could suggest that these molecules were potentially misclassified with other metabolites of similar structures.

Common metabolites were noted to be upregulated shortly after finishing concurrent chemoradiation therapy and later on after concluding treatment with adjuvant TMZ. One of these metabolites is N-methylisoleucine, which is an isoleucine derivative belonging to the organic acids super class, and an association between this metabolite and cancer has not previously been reported. Another metabolite that was upregulated is 4-methyl-5-thiazoleethanol which is a thiazole with an organoheterocyclic compound super class; this is a natural product that is found outside the cells as well as in the cytoplasm. The hydroxy fatty acid 6-hydroxycaproic acid was upregulated after concurrent chemoradiation.

Another set of metabolites was noted to be downregulated shortly after finishing concurrent chemoradiation therapy and later after concluding treatment with adjuvant TMZ. This was related to medication changes during treatment such as discontinuing famotidine (thiazole with an organoheterocyclic super class) after treatment. N-isovalerylglycine, an amino acid natural product of leucine catabolism [32], has been associated with multiple diseases including colorectal cancer, although no association has been established with brain tumors. Further, there has not been any reported association of cancer with 3-methylcrotonylglycine.

After concluding treatment, we noted an increase in coniferyl aldehyde, which is a cinnamic acid with the polyketide super class. This metabolite has been reported to promote the re-proliferation of the intestinal epithelium by inhibiting cell death and promoting endothelial cell function [33], though there has not been any reported association with brain tumors. Dimethylsulfoxide is probably due to an exogenous source given that it has analgesic and anti-inflammatory properties.

Glycerophosphocholine is an organic phosphoric acid with the organic acid super class. It has a role in choline storage in the cytosol. Glycerophosphocholine has been linked to cancer, with lung cancer tissue having higher levels than adjacent normal tissues [34]. Furthermore, melanoma with brain metastasis has higher levels of glycerophosphocholine [35].

The upregulation of bradykinin after concluding treatment could be of clinical value as bradykinin was previously reported to promote glioma invasion and cell migration by acting on B2 receptors [36]. The authors of this report propose targeting B2 receptors as a treatment strategy in the future. The upregulation of diatrizoic acid, which is an organic, iodinated radiopaque X-ray contrast medium used in diagnostic radiography, is likely due to imaging studies performed near sample collection times.

Treatment with temozolomide causes senescence in cancer cells [37], which might explain our observation of increased levels of carnitine during treatment. Carnitines have previously been reported to lead to senescence [38] wherein the downregulation of L-propionylcarnitine was seen after finishing treatment with chemotherapy. 1,5-pentanediamine is a diamine that has previously been reported to inhibit ornithine decarboxylase, which results in the inhibition of neuroblastoma cells and glioma cells to a lesser extent [39]. Chenodeoxycholic acid 24-acyl-beta-D-glucuronide, mostly an exogenous metabolite, is a bile acid with the sterol lipid super class.

In our study, combined treatment comprising surgery, radiation therapy, and chemotherapy led to notable alterations in the levels of endogenous and exogenous compounds. The changes in endogenous compounds, such as glycine and serine, are consistent with typical cancer-related metabolites, possibly resulting from perturbed glycolysis. However, it is plausible that some of these metabolites may represent hexoses, necessitating further validation. On the other hand, the complete significance of exogenous metabolites in relation to the treatment or treatment stage remains to be fully elucidated.

Furthermore, ML models, including random forest (RF) and AdaBoost (AB), accurately classified treatment phases, with RF showing the highest accuracy. The RF model successfully distinguished pre-surgery from post-surgery plasma samples, capturing treatment-related metabolic changes. Sorbitol and N-methylisoleucine emerged as important predictive features, consistently ranking highly across multiple methods. SHAP analysis confirmed their significance in treatment stage classification. Notably, sorbitol and N-methylisoleucine, previously unassociated with cancer, hold potential as novel glioblastoma biomarkers. The ML findings complemented statistical analyses, providing further insights into predictive biogenic amines. This integration enhances our understanding of glioblastoma metabolism.

The findings of this study are limited by the small sample size and it being a single-center study. An additional external dataset from multiple institutions would be needed to confirm our results. Second, the study is missing control groups consisting of samples from normal donors. However, comparing metabolites longitudinally in the same patient throughout treatment stages is considered, using every patient as their own control in a prospective fashion. Lastly, due to the small sample size, the machine learning models suffer from some degree of overfitting despite our efforts to prevent this. With a larger sample size, however, we believe our machine learning models will be more generalizable. Furthermore, with longitudinal data, we can further apply more sophisticated deep learning models such as RNN, LSTM [40], and transformer [41].

## 5. Conclusions

In this study of glioblastoma IDH *wild-type* patients, we observed notable changes in plasma metabolites associated with surgery, radiation, and chemotherapy. To ensure the robustness of these findings, further validation in a larger and more diverse patient group is necessary. Additionally, delving into the underlying mechanisms of these metabolites will provide a better understanding of their role. Such efforts could offer a new and practical approach for diagnosing and monitoring treatment responses, potentially benefiting patient care.

## Figures and Tables

**Figure 1 biomedicines-11-02261-f001:**
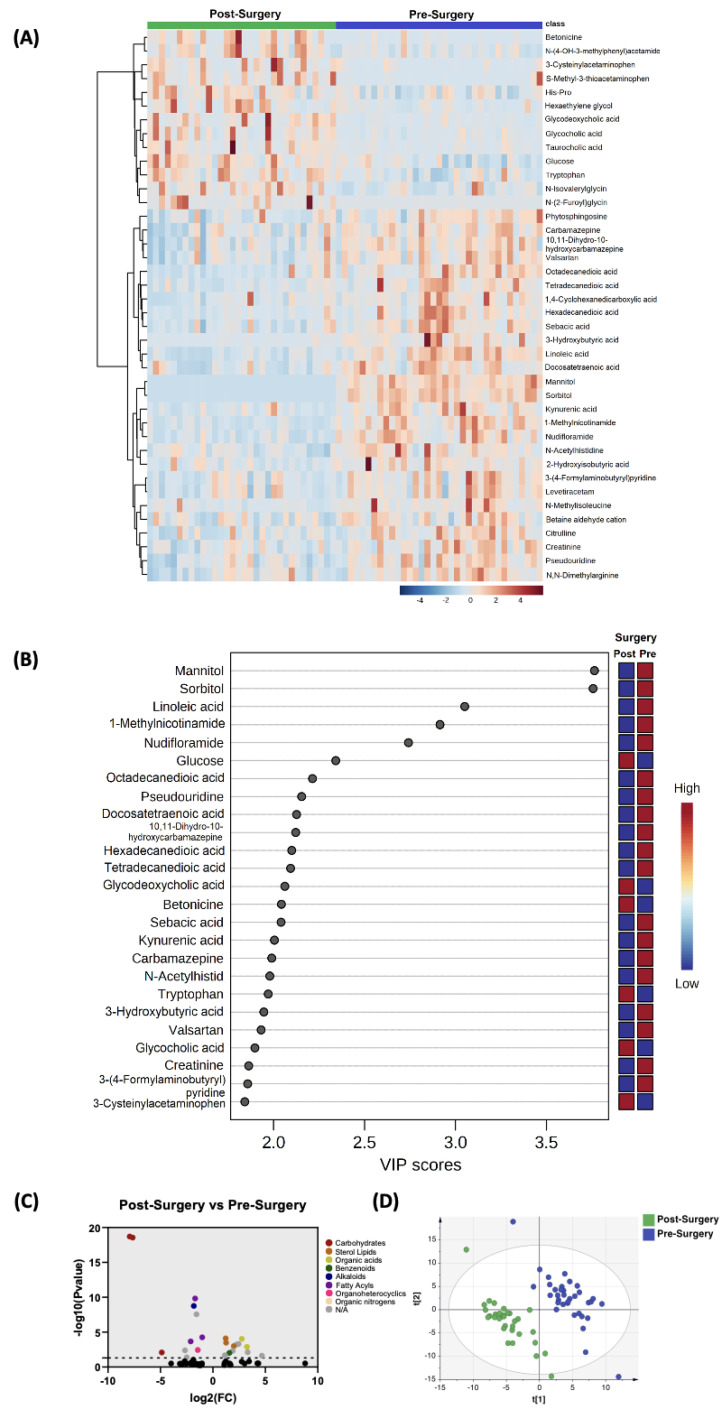
Comparison of biogenic amine profile of post-surgery samples vs. pre-surgery. (**A**) Partial least squares–discriminant analysis (PLS-DA) showed two-group clustering with evidence of separation between plasma at pre-surgery (purple) and post-surgery (green). (**B**) VIP score with the highest 25 metabolites. (**C**) Heat map of the top 40 altered metabolites pre-surgery and post-surgery. Blue indicates decreased peak value and maroon indicates increased peak value of each compound listed. (**D**) Volcano plot of upregulated biogenic amine (right side) and downregulated biogenic amine (left side) plasma specimens post-surgery compared to pre-surgery using *p*-value of <0.05 and fold change cutoffs of 2.0; colors correlate with metabolite’s super class.

**Figure 2 biomedicines-11-02261-f002:**
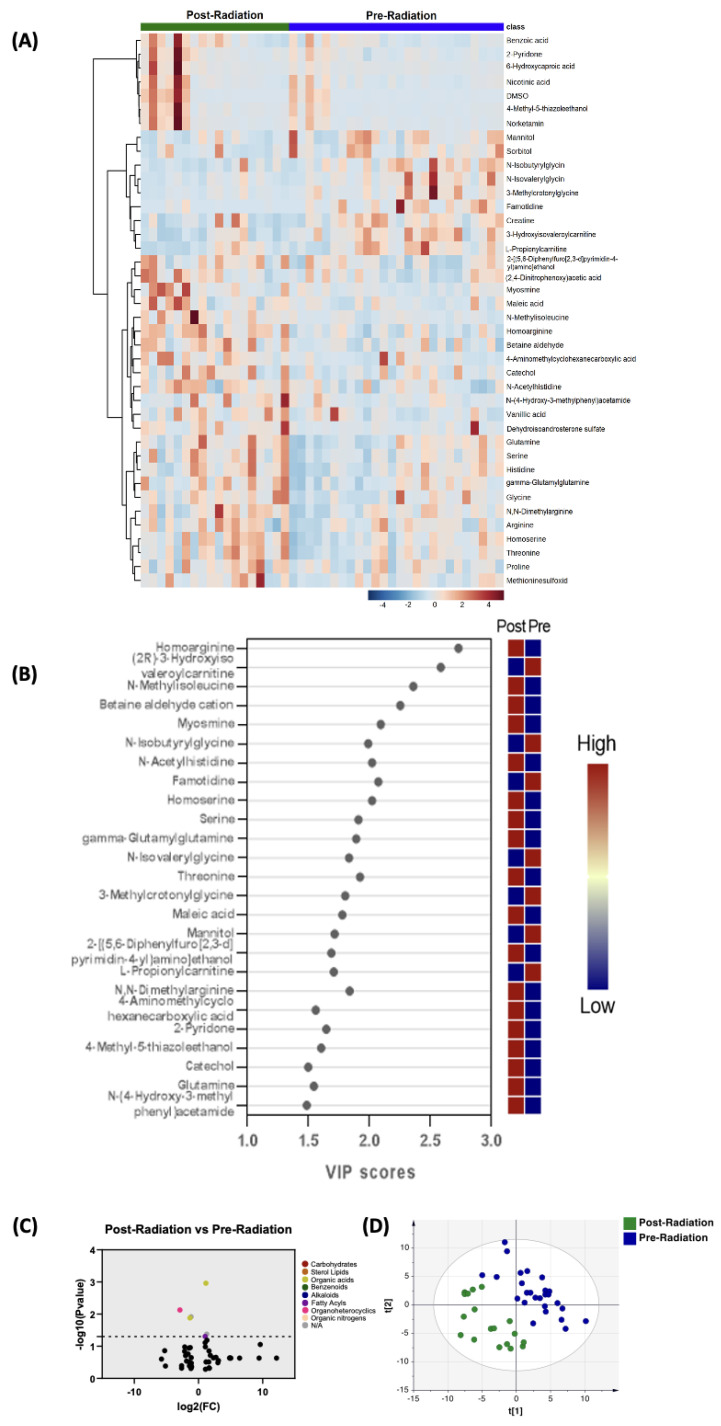
Comparison of biogenic amines profile of post-radiation samples vs. pre-radiation. (**A**) Partial least squares–discriminant analysis (PLS-DA) showed two-group clustering with evidence of separation between plasma at pre-radiation (purple) and post-radiation (green). (**B**) VIP score with the highest 25 metabolites. (**C**) Heat map of the top 40 altered metabolites pre-radiation and post-radiation. Blue indicates decreased peak value and maroon indicates increased peak value of each compound listed. (**D**) Volcano plot of upregulated biogenic amines (right side) and downregulated biogenic amines (left side) in plasma specimens post-radiation compared to pre-radiation using *p*-value of <0.05 and fold change cutoffs of 2.0; colors correlate with metabolite’s super class.

**Figure 3 biomedicines-11-02261-f003:**
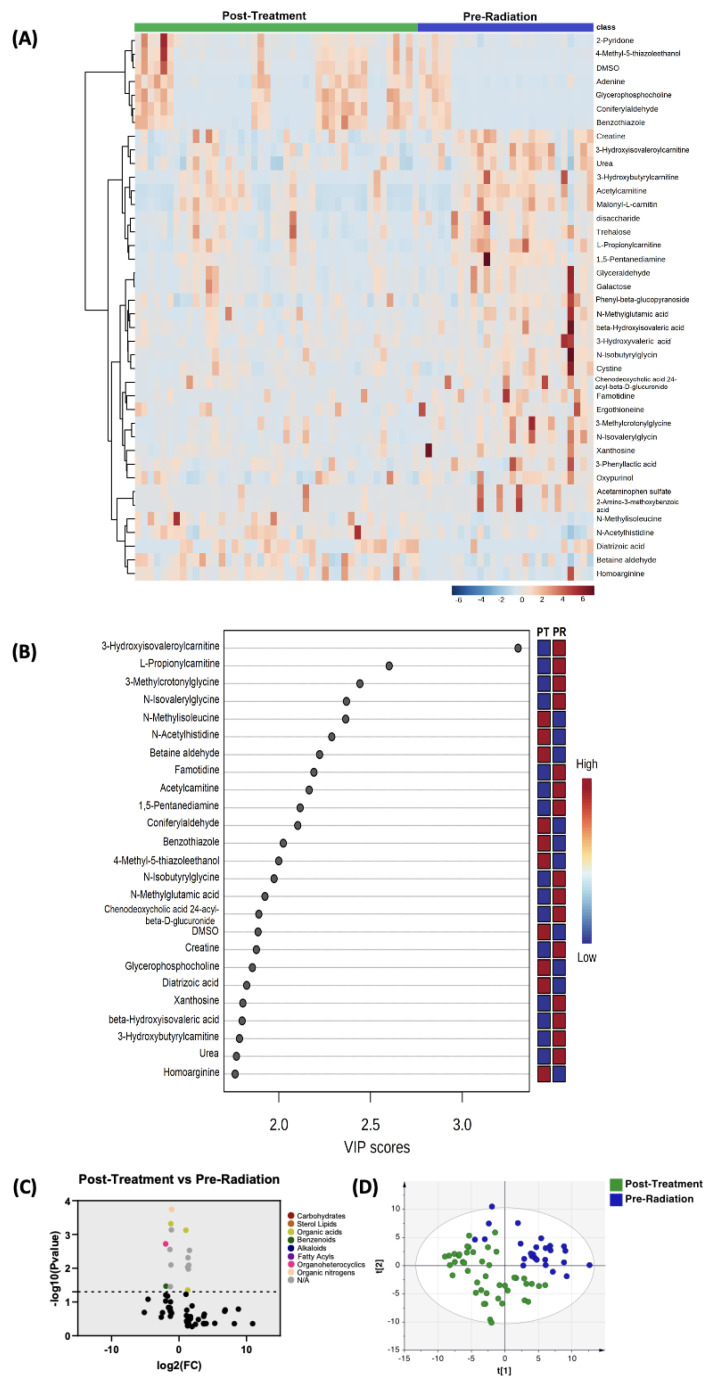
Comparison of biogenic amine profiles of post-treatment samples vs. pre-radiation samples. (**A**) Partial least squares–discriminant analysis (PLS-DA) showed two-group clustering with evidence of separation between plasma pre-radiation (purple) and post-treatment (green). (**B**) VIP score with the highest 25 metabolites. (**C**) Heat map of the top 40 altered metabolites pre-radiation and post-radiation. Blue indicates decreased peak value and maroon indicates increased peak value of each compound listed. (**D**) Volcano plot of upregulated biogenic amines (right side) and downregulated biogenic amines (left side) in plasma specimens post-treatment compared to pre-radiation using *p*-value of <0.05 and fold change cutoffs of 2.0; colors correlate with metabolite’s super class.

**Figure 4 biomedicines-11-02261-f004:**
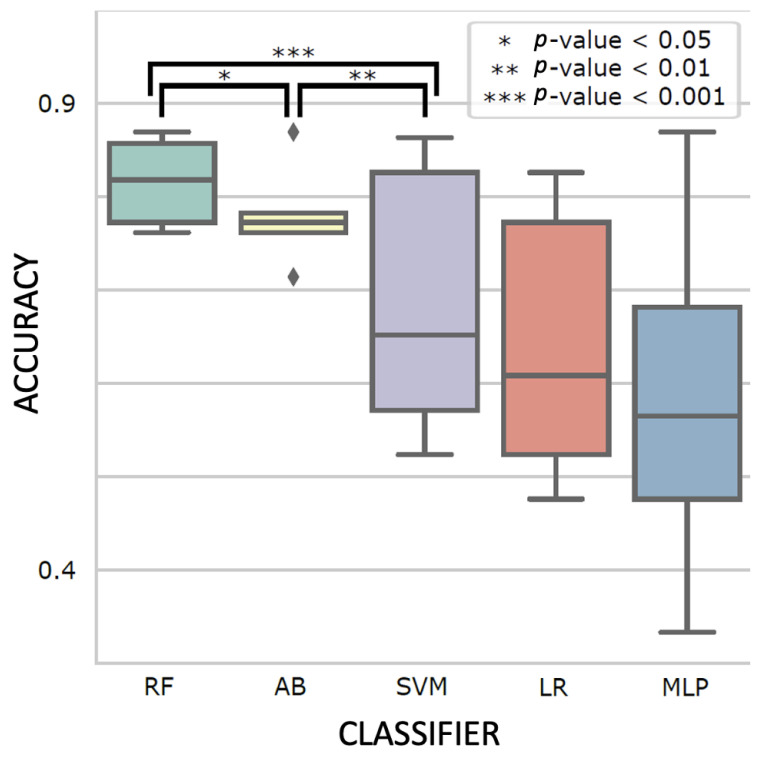
Comparison between random forest (RF), AdaBoost (AB), Support Vector Machine (SVM), Logistic Regression (LR), and Multiplayer Perceptron (MLP) classifiers on 5-fold cross-validation results. Five independent grid searches were performed for robustness, and the hyperparameter set with the best mean validation accuracy was shown for each classifier (i.e., *n* = 25 for each boxplot). The box indicates the interquartile range, the horizontal middle line indicates the median, the diamond indicates an outlier, and the whisker line indicates the range between minimum and maximum, excluding outliers. The *p*-values were computed with two-tailed *t*-tests.

**Figure 5 biomedicines-11-02261-f005:**
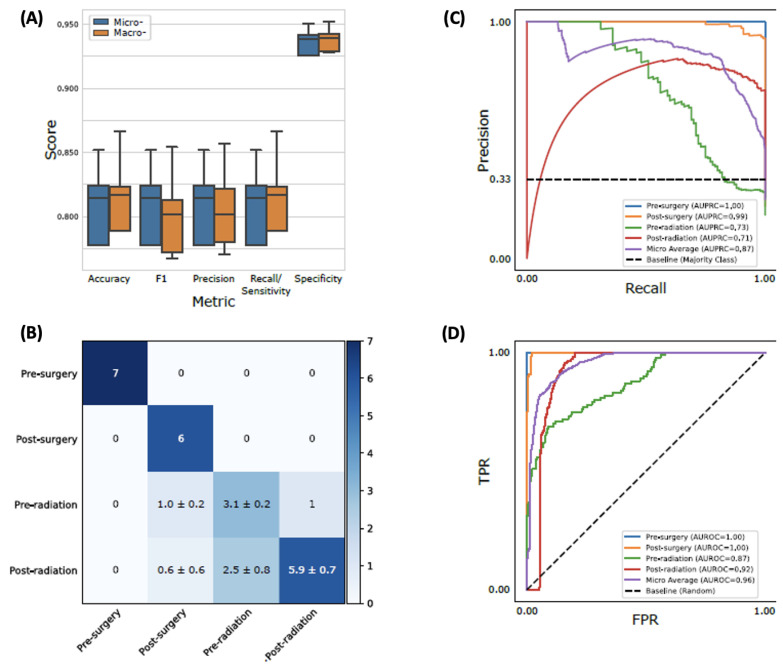
Performance evaluation on the testing set for the best model. Random forest with the best hyperparameter trained with the entire training set was shown. (**A**) Micro- and macro-averaging of evaluation metrics. Macro-averaging is the average of a metric across different classes, while micro-averaging is the weighted average based on the size of each class. The boxplot consists of 20 points, where each point is a corresponding metric computed from an independent run with different random initialization. (**B**) Confusion matrix for the best model. The standard deviation was computed from the 20 independent runs. (**C**,**D**) Precision–recall (PR) and receiver operating characteristic (ROC) curves. Each curve was plotted by concatenating predictions from the 20 independent runs.

**Figure 6 biomedicines-11-02261-f006:**
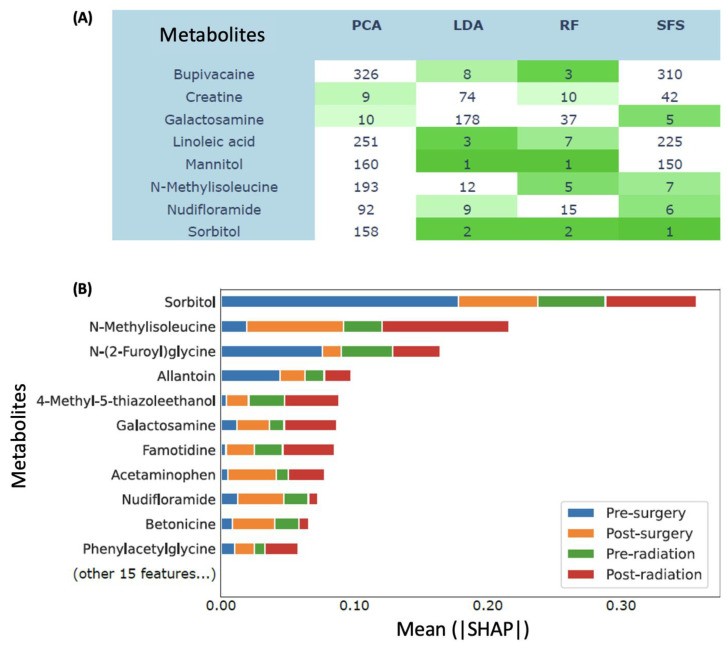
Comparison of different feature importance methods. (**A**) Metabolite feature ranks of Principal Component Analysis (PCA), Linear Discriminant Analysis (LDA), random forest (RF), and sequential feature selection (SFS). The table only shows the biogenic amines ranked top 10 by at least two methods. Dark green represents tope 3 ranked metabolites, and light green represents top 10 ranked metabolites (**B**) SHAP values of random forest with features selected via SFS. The interpretation of a SHAP value for a tree-based model is that a feature satisfying a split criterion expects to change the output probability by the amount of SHAP value.

**Table 1 biomedicines-11-02261-t001:** Demographics of patients. “BS”, pre-surgery sample; “S”, two days post-surgery sample; “RT”, pre-radiation sample; and “PRT”, immediate post-radiation sample; “PT”, post-treatment sample. “X” indicates that a sample was collected at the listed time point for each patient.

Patient ID	Sex	Ethnicity	Diagnosis Age (Years)	BMI at Diagnosis	BS	S	RT	PRT	PT
**1**	M	White	60	40	X	X	X	X	X
**2**	M	White	72	30	X	X	X		
**3**	M	Hispanic	43	28	X	X		X	X
**4**	M	Asian	49	57	X	X		X	
**5**	F	White	78	23	X	X			
**6**	M	Hispanic	65	22	X	X		X	X
**7**	M	White	72	41	X	X	X		
**8**	M	White	80	24	X	X	X	X	
**9**	F	White	61	27	X	X	X		
**10**	F	White	69	25	X	X	X		
**11**	M	Indian	60	27	X		X	X	X
**12**	F	White	61	25	X	X	X		
**13**	F	White	52	27	X	X			
**14**	M	White	62	30	X	X	X		
**15**	M	White	69	31	X	X	X	X	X
**16**	M	White	67	44	X	X			
**17**	F	White	82	28	X	X	X		
**18**	F	White	55	29	X	X			
**19**	M	African American	47	37	X	X	X	X	
**20**	M	White	63	30	X	X	X	X	X
**21**	F	White	86	27	X	X	X		
**22**	F	White	64	31	X	X	X	X	
**23**	M	White	56	22	X	X	X	X	
**24**	F	White	69	26	X	X	X	X	X
**25**	F	NA	69	27	X	X	X		X
**26**	M	White	64	36	X	X	X	X	X
**27**	M	White	68	28	X		X	X	
**28**	M	White	69	28	X	X	X	X	X
**29**	F	White	58	27	X		X	X	
**30**	F	white	66	27	X	X	X		
**31**	M	White	55	28	X	X	X	X	X
**32**	F	White	60	20	X	X	X		X
**33**	M	White	58	28	X	X	X		
**34**	M	White	53	30	X	X	X		X
**35**	M	White	58	26	X	X	X		
**36**	M	White	76	35	X				

**Table 2 biomedicines-11-02261-t002:** List of metabolites that were upregulated or downregulated in the plasma of patients with glioblastoma after surgery as compared with prior to surgery.

Upregulated Metabolites	*p* Value	Downregulated Metabolites	*p* Value
glycodeoxycholic acid	7.79 × 10^−5^	Mannitol	1.96 × 10^−19^
Betonicine	9.23 × 10^−5^	Sorbitol	2.75 × 10^−19^
Glycocholic acid	3.18 × 10^−3^	Linoleic acid	1.55 × 10^−10^
3-cysteinylacetaminophen	4.95 × 10^−3^	1-methylnicotinamide	1.78 × 10^−9^
S-methyl-3-thioacetaminophen	5.77 × 10^−3^	Nudifloramide	2.81 × 10^−8^
Taurocholic acid	9.97 × 10^−3^	Hexadecanedioic acid	5.49 × 10^−5^
Glycine	0.0013	3-hydroxybutryic acid	2 × 10^−3^
p-acetamidophenyl-beta-D-glucuronide	0.0048	Riluzole	0.0036
2-hydroxy-5-sulfopyridine-3-carboxylic acid	0.0074	Bupivacaine	0.0041
Acetaminophen sulfate	0.0095	Lactitol	0.008
2-amino-3-methoxybenzoic acid	0.02	1-hydroxymidazolam-beta-D-glucuronide	0.041
dehydrofelodipine	0.024		

**Table 3 biomedicines-11-02261-t003:** List of metabolites that were upregulated or downregulated in the plasma of patients with glioblastoma after radiation as compared with prior to radiation.

Upregulated Metabolites	*p* Value	Downregulated Metabolites	*p* Value
N-methylisoleucine	0.0011	Famotidine	0.0074
4-methyl-5-thiazoleethanol	0.042	N-isovalerylglycine	0.012
6-hydroxycaproic acid	0.049	methylcrotonylglycine	0.0131

**Table 4 biomedicines-11-02261-t004:** List of metabolites that were upregulated or downregulated in the plasma of patients with glioblastoma after adjuvant chemoradiation as compared with prior to radiation.

Upregulated Metabolites	*p* Value	Downregulated Metabolites	*p* Value
N-methylisoleucine	8 × 10^−4^	L-propionylcarnitine	2 × 10^−4^
coniferylaldehyde	0.003	3-methylcrotonylglycine	5 × 10^−4^
4-methyl-5-thiazoleethanol	0.0049	N-isovalerylglycine	7 × 10^−4^
dimethylsulfoxide	0.0081	famotidine	0.0019
glycerophosphocholine	0.0093	1,5-pentanediamine	0.0028
Diatrizoic acid	0.011	Chenodeoxycholic acid 24-acyl-beta-D-glucuronide	0.008
bradykinin	0.044	Acetaminophne sulfate	0.035
		2-amino-3-methoxybenzoic acid	0.035

## Data Availability

Data are not publicly available due to privacy reasons.

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
