# Peer review of "Profile Characterization of Biogenic Amines in Glioblastoma Patients Undergoing Standard-of-Care Treatment"

_biomedicines, 2023, doi:10.3390/biomedicines11082261_

Round 1

Reviewer 1 Report

In the submitted manuscript there are some both editorial and very important substantive errors that should be corrected before proceeding further. Honestly, I was hesitating between major revision and rejection.

Abstract is over 300 words and the limit is 200. Please shorten it. For example, in Lines 27-41 a lot of technical information is provided. Honestly, at this point, the reader is not interested in those values when stated without the broader spectrum beforehand. Therefore, I suggest pointing out only the most important findings.

References should be formatted like [1, 2] and not [Surname, year].

In the introduction, the authors should mention the similar approaches (biogenic amines profile characterization) in other types of cancer.

Line 67, what about alternating electric field therapy?

Line 165, full list of this amines should be included in supplementary information

Lines 177-178, the values should be presented solely in a form of a table as presenting the results this way decreases the clarity

Figure 1B, this is quite confusing. I understand the presence of some biogenic amines on this list (i.e. creatinine, tryptophan), but besides them there are also some compounds that are not amines (i.e. glucose) or drug and their metabolites (i.e. carbamazepine). For me it’s clear that carbamazepine would be detected only in patients taking this API.

Lines 250, AUPRC and AUROC are not defined

Line 266, sorbitol is not amine at all!

Line 270, linoleic acid, mannitol, are not amines. I strongly recommend consulting the results with someone with the knowledge of organic chemistry or biochemistry.

Line 313, there’s no such thing as N-glycine…

Lines 318-319, other places too, why Glycine is with capital G? It also applies to other compounds mentioned in this work.

Lines 352-354, therefore, it is highly probable that those compounds were not identified correctly. There are multiple compounds with the same chemical formula.

Figure 6, again, a lot of compounds that are not amines (i.e. sorbitol) and a lot of drugs

Line 572, please remove „37.”

Mentioned above.

Author Response

Please see our responses to Reviewer 1 in the attached document. Thank you for your time and consideration.

Reviewer 2 Report

Dear authors,

 You have done a great work in the search for molecular predictors in the therapy of glioblastoma. To my point of view there are several aspects that need explanations:

1. A definition of biogenic amines is lacking. Despite the lack of a global accepted definition of biogenic amines, according to Akyol et al, they can be defined as substances produced by decarboxylation of an amino acid in a one-step reaction or a in a series of reactions and with an important physiological effect. (Akyol O, Tessier K, Akyol S. Accuracy and uniformity of the nomenclature of biogenic amines and polyamines in metabolomics studies: A preliminary study. Biochem Mol Biol Educ. 2021; 49:441–445. https://doi.org/10.1002/bmb.21497).

2. A description of the implications of biogenic amines in the biology of glioblastoma is needed.

3. A research hypothesis explaining what are the researchers looking for is also needed.

4. To my point of view a control group of patients with operations for non-tumour conditions is needed, to define which changes in biogenic amines are related with the surgical process.

5. It should be revised if the described biogenic amines fulfil the criteria of its definition.

Author Response

(The authors gave the same response as above.)

Round 2

Reviewer 1 Report

The Authors have improved and corrected their manuscript. Current version can be accepted.

Reviewer 2 Report

Dear authors,

The manuscript can be accepted in the present form.